# Speaking Guided by Listening: Unsupervised Text-to-Speech Generative Model Guided by End-to-End Speech Recognition

## Abstract

We propose to utilize end-to-end automatic speech recognition (E2EASR) as a guidance model to realize unsupervised text-to-speech (TTS). An unconditional score-based generative model (SGM) is trained with untranscribed speech data. In the sampling stage, the unconditional score estimated by the SGM is combined with the gradients from ASR models by the Bayes rule to get the conditional score. We use a set of small ASR models trained only on 80-hour labeled ASR data to guide the unconditional SGM and generate speech with high-quality scores in both objective and subjective evaluation. Similarly, we can also use additional speaker verification models to control speaker identity for the synthesized speech. That allows us to do the zero-shot TTS for the target speaker with a few seconds of enrollment speech. Our best unsupervised synthesized speech gets $\sim 8\%$ word error rate in testing, and the best speaker-controlled TTS gets 3.3 mean opinion score (MOS) in the speaker similarly testing.

## 1 Introduction

Text-to-speech (TTS) systems have made significant progress due to the need for natural, expressive speech in various applications such as virtual assistants, audiobooks, and automated customer service. With the rise of neural network-based models, TTS has transformed from traditional concatenative and parametric methods to more advanced deep learning approaches, leading to significant improvements in quality. (Ren et al., 2019a; Li et al., 2019; Shen et al., 2018). The adoption of autoregressive models like Tacotron and Transformer TTS (Wang et al., 2017; Li et al., 2019) has allowed for significant improvements in the synthesis of natural-sounding speech. The emergence of non-autoregressive models, such as FastSpeech (Ren et al., 2019a; 2020), further expanded the capabilities and inference speed of TTS by introducing parallel speech generation techniques. Moreover, diffusion models Sohl-Dickstein et al. (2015); Ho et al. (2020); Song & Ermon (2019), which have shown high performance in image generation tasks, have recently been adapted for speech synthesis with promising results (Jeong et al., 2021; Popov et al., 2021; Huang et al., 2022; Tae et al., 2022). Despite these advancements, mainstream TTS models rely heavily on high-quality paired text-speech data. This dependence remains a critical barrier to developing robust TTS systems for languages and speakers with limited available data.

In this paper, we propose an E2EASR-guided method for unsupervised TTS [1]. We trained an unconditional score-based generative model (SGM) on the 522-hour LibriTTS-R (Koizumi et al., 2023) dataset without using any text or speaker label. Only 80-hour WSJ (Garofolo, John S. et al., 2007) labeled ASR data is used for training the guidance ASR models. ASR data is usually much more accessible to collect than TTS data. In fact, the ASR system can be directly trained on noisy or low-quality speech, while the TTS usually requires studio-quality recording. In addition, many of the recent TTS works still need phoneme-level supervision to learn the phoneme duration (Ammar Abbas et al., 2022; Effendi et al., 2022; Kim et al., 2020; 2022b), while the majority of ASR systems, in recent years, are simply trained in end-to-end style with sentence-level annotations (Graves et al., 2013; Bahdanau et al., 2016; Kim et al., 2017). We propose to use multiple independent ASR sys-

---

[1]Demos are available at `https://asr-guided-tts.github.io/` (Anonymous version)

tems trained on the same dataset to provide more robust joint guidance. This allows us to generate higher-quality speech than what is possible by using guidance from only a single-ASR system.

We also propose using a speaker verification model for zero-shot target speaker TTS, in which only a few seconds of enrollment speech to generate speech for unseen speakers. By assuming the two statistically independent and using the Bayes rule we can combine the guidance from ASR and speaker models without conflicts or scale issues. The two can thus jointly guide the unconditional SGM to realize target speaker- and text-conditioned diffusion generation.

Another benefit of the E2EASR guidance is that the duration model in conventional TTS, which estimates the phoneme duration, can be omitted. Our experiments show that we can generate high-quality speech with the input of target text and a total desired speech length. The end-to-end training of ASR models enables them to implicitly handle phoneme duration, helping the SGM generate speech at a natural pace.

## 2 BACKGROUND

### 2.1 UNCONDITIONAL DIFFUSION FOR SPEECH GENERATION

We follow the unconditional generation model (SGM) introduced by Song et al. (2021). It unifies the denoising score matching with Langevin dynamics (SMLD) (Song & Ermon, 2019) and denoising diffusion probabilistic model (DDPM) (Ho et al., 2020) frameworks by using stochastic differential equations (SDEs). For speech generation, we model the speech data in the mel-spectrogram domain $\mathbf{X} \in \mathbb{R}^{L \times F}$, where $L$ is the length of the spectrum and $F$ is the number of frequency banks. Let $p_0(\mathbf{X})$ be the probability density function (p.d.f.) of the clean speech data. The SDE describes the forward diffusion process, which converts the distribution $p_0(\mathbf{X})$ to a simple prior distribution $p_T(\mathbf{X})$ by a continuous time variable $t \in [0, T]$:

$$d\mathbf{X} = \mathbf{F}(\mathbf{X}, t)dt + g(t)d\mathbf{W}, \tag{1}$$

where $dt$ is an infinitesimal timestep , $\mathbf{W} \in \mathbb{R}^{L \times F}$ is a Brownian motion, $\mathbf{F}(\cdot)$ is a matrix-valued *drift* function, and $g(t)$ is the scalar value *diffusion* coefficient determined by $t$. $\mathbf{F}(\cdot)$ is the deterministic part of the SDE, while $g(t)$ controls the scale of the noise-adding process. Based on previous research from Anderson (1982), there is a reverse SDE that describes the reverse diffusion process corresponding to the above forward process:

$$d\mathbf{X} = [-\mathbf{F}(\mathbf{X}, t) + g(t)^2 \nabla_{\mathbf{X}} \log p_t(\mathbf{X})]dt + g(t)d\bar{\mathbf{W}}, \tag{2}$$

where $\bar{\mathbf{W}}$ is the Brownian motion in the reverse process, $dt$ is a negative infinitesimal timestep, $\nabla_{\mathbf{X}} \log p_t(\mathbf{X})$ is the gradient of the logarithm data distribution $p_t(\mathbf{X})$ at timestep $t$, i.e., the *score* of the distribution. We can train a model to approximate such a score function, i.e., a score model $s_{\boldsymbol{\theta}}(\mathbf{X}, t)$ parameterized by $\boldsymbol{\theta}$ to estimate $\nabla_{\mathbf{X}} \log p_t(\mathbf{X})$ by using the score matching (Hyvärinen & Dayan, 2005; Song & Ermon, 2019) method:

$$\boldsymbol{\theta}^* = \arg\min_{\boldsymbol{\theta}} \mathbb{E}_{t, \mathbf{X}_0, \mathbf{X}_t | \mathbf{X}_0} \left[ \|\mathbf{s}_{\boldsymbol{\theta}}(\mathbf{X}_t, t) - \nabla_{\mathbf{X}_t} \log p_{0t}(\mathbf{X}_t | \mathbf{X}_0)\|^2 \right], \tag{3}$$

where $\mathbf{X}_0 \sim p_0(\mathbf{X})$ is the clean training data and $\mathbf{X}_t \sim p_{0t}(\mathbf{X}_t | \mathbf{X}_0)$ is the perturbed data sampled from the conditional distribution $p_{0t}(\mathbf{X}_t | \mathbf{X}_0)$ at timestep $t$.

After the score model has been trained, $\nabla_{\mathbf{X}} \log p_t(\mathbf{X})$ in Eq.2 can be replaced with $s_{\boldsymbol{\theta}}(\mathbf{X}, t)$ for inference. We can start from the prior distribution $\mathbf{X}_T \sim p_T(\mathbf{X})$, and generate samples from the target distribution by using a numerical SDE solver to solve the reverse SDE. Commonly used numerical solvers include the Euler-Maruyama method and predictor-corrector sampler (Song et al., 2021).

### 2.2 CLASSIFIER GUIDANCE

Classifier guidance (Song et al., 2021; Ho et al., 2020) can generate desired data by using the unconditional diffusion model and an external classifier. To generate the class conditioned data, Eq.2 can be updated into the following conditioned form:

$$d\mathbf{X} = [-\mathbf{F}(\mathbf{X}, t) + g(t)^2 \nabla_{\mathbf{X}} \log p_t(\mathbf{X}|y)]dt + g(t)d\bar{\mathbf{W}}, \tag{4}$$

where $p_t(\mathbf{X}|y)$ is the data p.d.f conditioned by discrete class $y$. According to the Bayes rule, $\nabla_{\mathbf{X}} \log p_t(\mathbf{X}|y)$ can be rewritten as:

$$\nabla_{\mathbf{X}} \log p_t(\mathbf{X}|y) = \nabla_{\mathbf{X}} \log p_t(\mathbf{X}) + \nabla_{\mathbf{X}} \log P_t(y|\mathbf{X}), \qquad (5)$$

The first term in Eq.5 can be estimated by the unconditional SGM $s_{\boldsymbol{\theta}}(\mathbf{X}, t)$, and the second term in Eq.5 can be estimated by the external classifier $P_{\boldsymbol{\phi}}(y|\mathbf{X})$ parameterized by $\boldsymbol{\phi}$. The labeled data used to train the classifier can also be different from the data used for the unconditional SGM. This allows a TTS system to be built without transcribed TTS data but only with ASR data. The latter is more accessible to collect since ASR does not require high-quality recordings as TTS does and thus can be scaled more easily.

## 3    RELATED WORKS

Recently, many diffusion-based TTS models have proven to be remarkably successful in the text-to-speech task (TTS) (Jeong et al., 2021; Popov et al., 2021; Huang et al., 2022; Tae et al., 2022). This emerging class of generative models adopts an iterative generative approach, where, during training, a complex data distribution is gradually corrupted by Gaussian noise (Song et al., 2021). These models are trained to estimate the gradient fields that reverse this process from a noisy prior, guiding the data sample back to its original distribution. And most of those diffusion models are conditioned on the text or semantic tokens. To train such models conditioned by the input text, large scale of high-quality paired text-speech data are required, posing a major practical problem for TTS (Ren et al., 2019b).

A recent work named Guided-TTS (Kim et al., 2022a) uses the classifier guidance method for TTS by using a phoneme classifier as the guidance model. Guided-TTS is the most related work to this paper. The main difference between Guided-TTS and this paper includes:

1) Guided-TTS needs to train a frame-level phoneme classifier, an additional duration predictor is required to estimate the duration of each phoneme, and the diffusion model is trained on single-speaker datasets. They use large-scale 960-hour labeled ASR data to train a phone-level classifier. while we only use 80-hour labeled ASR data to train end-to-end ASR guidance models.

2) Guided-TTS trains unconditional diffusion models on single-speaker training datasets. It can not synthesize the target speaker differently from the training speaker. A follow-up work named Guided-TTS2 (Kim et al., 2022b) replaced the unconditional diffusion model with a speaker-conditioned diffusion model to make it able for speaker-conditional TTS. That requires the diffusion training data to be labeled for the speaker. In this work, we proposed to use speaker verification models for TTS guidance. That allows us to do the zero-shot target speaker TTS with a few seconds of enrollment speech. We do not need any speaker labels for the diffusion model training.

3) In the Guided-TTS, the authors use norm-based gradient scaling methods to combine the gradients from the unconditional diffusion model and the guidance classifier. In this paper, we follow Eq.5 to combine $\nabla_{\mathbf{X}} \log p_t(\mathbf{X})$ and $\nabla_{\mathbf{X}} \log P_t(y|\mathbf{X})$ without any scaling weight that hurt the Bayes rule. The impact of guidance gradients is instead tuned by the temperature in the Softmax function when estimating $P_t(y|\mathbf{X})$. Following the Bayes rule allows us to combine multiple guidance models more easily and safely by making an independent assumption between the guidance models.

## 4    TEXT-TO-SPEECH GUIDED BY END-TO-END ASR

In this section, we propose using E2EASR as guidance for unsupervised TTS. Our method involves three different modules: 1) an unconditional score-based generative model, which models the distribution of clean speech from the speech data without transcription; 2) E2EASR systems that provide the gradient guidance conditioned by the target text; 3) optional speaker verification models for controlling the target speaker identity, enabling zero-shot target speaker TTS with only a few seconds of reference speech.

### 4.1    UNCONDITIONAL SCORE-BASED GENERATIVE MODEL GUIDED BY E2EASR

Here, we introduce the E2EASR guidance mechanism that enables TTS with the unconditional SGM. We denote the one-hot text tokens as $\mathbf{Y} \in \{0,1\}^{K \times V}$, where $K$ is the length of the text and

$V$ is the vocabulary size. The TTS task by score-based modeling is to model the distribution of $p(\mathbf{X}|\mathbf{Y})$, where $\mathbf{X} \in \mathbb{R}^{L \times F}$ is the mel-spectrogram with length[2] $L$ and the number of frequency banks $F$. We train the unconditional SGM with the objective function in Eq.3. To generate the speech conditioned by $\mathbf{Y}$, the reverse SDE can be written as:

$$d\mathbf{X} = [-\mathbf{F}(\mathbf{X}, t) + g(t)^2 \nabla_{\mathbf{X}} \log p_t(\mathbf{X}|\mathbf{Y})]dt + g(t)d\bar{\mathbf{W}}, \tag{6}$$

similar to that discussed in Sec.2.2, $\nabla_{\mathbf{X}} \log p_t(\mathbf{X}|\mathbf{Y})$ can be decomposed into the sum of $\nabla_{\mathbf{X}} \log p_t(\mathbf{X})$ and $\nabla_{\mathbf{X}} \log P_t(\mathbf{Y}|\mathbf{X})$: The former gradient can be estimated by the unconditional SGM $s_\theta(\mathbf{X}, t)$, while the latter gradient can be calculated by differentiating the ASR models, which were trained to estimate $P_t(\mathbf{Y}|\mathbf{X})$.

We use the joint CTC-Attention (Kim et al., 2017; Watanabe et al., 2018) E2EASR, which combines the connectionist temporal classification (CTC) (Graves et al., 2006; 2013) and Attention-based Encoder-Decoder (AED) (Chan et al., 2016; Bahdanau et al., 2016). With the joint CTC-Attention ASR, $P_t(\mathbf{Y}|\mathbf{X})$ can be jointly represented as:

$$P_t(\mathbf{Y}|\mathbf{X}) = P_t(Y_{CTC} = \mathbf{Y}, Y_{AED} = \mathbf{Y}|\mathbf{X}), \tag{7}$$

If we assume the CTC task and AED components estimate the target text $\mathbf{Y}$ independently in their different feature spaces, then:

$$\nabla_{\mathbf{X}} \log P_t(\mathbf{Y}|\mathbf{X}) = \nabla_{\mathbf{X}} \log P_t^{CTC}(\mathbf{Y}|\mathbf{X}) + \nabla_{\mathbf{X}} \log P_t^{AED}(\mathbf{Y}|\mathbf{X}). \tag{8}$$

We train the ASR model with perturbed data $\mathbf{X}_t$ and text label $\mathbf{Y}$ to make sure that the model can estimate a reliable $P_t(\mathbf{Y}|\mathbf{X})$ at each SDE timestamp $t$.

So far, with the above-proposed method it should be theoretically possible to control the generated speech from the unconditional SGM. However, the mel-spectrogram space $\mathbb{R}^{L \times F}$ is sparse, and in the sampling stage, the numerical SDE solver tends to sample a locally optimal $\hat{\mathbf{X}}_0$ leading to high log-likelihood with guidance ASR but poor quality. To alleviate this problem, we propose joint guidance by multiple ASR systems. We train $N$ compact ASR systems with a small number of parameters ($\approx 15\,\mathrm{M}$) using the same perturbed training data and assume that each ASR system is independent of the others when estimating $\mathbf{Y}$, then the 8 can be rewritten as:

$$\nabla_{\mathbf{X}} \log P_t(\mathbf{Y}|\mathbf{X}) = \sum_{n=1}^{N} (\nabla_{\mathbf{X}} \log P_t^{CTC_n}(\mathbf{Y}|\mathbf{X}) + \nabla_{\mathbf{X}} \log P_t^{AED_n}(\mathbf{Y}|\mathbf{X})), \tag{9}$$

where $P_t^{CTC_n}(\mathbf{Y}|\mathbf{X})$ and $P_t^{CTC_n}(\mathbf{Y}|\mathbf{X})$ are the CTC-ASR and AED-ASR the $n$-th joint E2EASR model, respectively.

## 4.2 SPEAKER CONDITIONAL GUIDANCE FOR ZERO-SHOT TTS

If speaker-id conditioning is added, the reverse SDE becomes:

$$d\mathbf{X} = [-\mathbf{F}(\mathbf{X}, t) + g(t)^2 \nabla_{\mathbf{X}} \log p_t(\mathbf{X}|\mathbf{Y}, c)]dt + g(t)d\bar{\mathbf{W}}, \tag{10}$$

where $c$ is the class of the target speaker, $\nabla_{\mathbf{X}} \log p_t(\mathbf{X}|\mathbf{Y}, c)$ is the score conditioned by target speaker and text. Given that $c$ and $\mathbf{Y}$ are conditionally independent, the conditioned score can be rewritten as:

$$\nabla_{\mathbf{X}} \log p_t(\mathbf{X}|\mathbf{Y}, c) = \nabla_{\mathbf{X}} \log p_t(\mathbf{X}) + \nabla_{\mathbf{X}} \log P_t(\mathbf{Y}, c|\mathbf{X}), \tag{11}$$

$$= \nabla_{\mathbf{X}} \log p_t(\mathbf{X}) + \nabla_{\mathbf{X}} \log P_t(\mathbf{Y}|\mathbf{X}) + \nabla_{\mathbf{X}} \log P_t(c|\mathbf{X}), \tag{12}$$

where $P_t(c|\mathbf{X})$ can be estimated by a speaker verification model $\mathbf{m}_{\boldsymbol{\mu}}$, where $\boldsymbol{\mu}$ is the pretained parameters. In the zero-shot target speaker TTS scenario, we can estimate $P_t(c|\mathbf{X})$ with the following method:

$$\mathbf{s}^c = \mathbf{m}_{\boldsymbol{\mu}}(\mathbf{X}^c), \tag{13}$$

$$\mathbf{s}_t = \mathbf{m}_{\boldsymbol{\mu}}(\hat{\mathbf{X}}_t), \tag{14}$$

$$P_t(c|X = \hat{\mathbf{X}}_t) \approx \frac{e^{\beta d(\mathbf{s}^c, \mathbf{s}_t)}}{e^{\beta d(\mathbf{s}^c, \mathbf{s}_t)} + \sum_i^M e^{\beta d(\mathbf{s}^i, \mathbf{s}_t)}}, \tag{15}$$

---

[2]The proposed method does not rely on a duration model. In the inference stage, $L$ can be empirically set or estimated by some simple algorithm based on the text length $K$.

where $\mathbf{X}^c$ is an enrollment utterance from target speaker $c$, $\mathbf{s}^c$ is the enrollment speaker vector extracted from $\mathbf{X}^c$; $\hat{\mathbf{X}}_t$ is the data sampled at current timestep $t$ in the diffusion reverse process, $\mathbf{s}_t$ is the current speaker vector extracted by the speaker model; scalar $\beta$ is a manually set parameter[3] which controls the sharpness of the distribution, whose inverse is also known as *temperature*. By tuning $\beta$, we can tune the impact of guidance gradients; function $d$ is a metric that measures the similarity of two vector embeddings, which is cosine similarity in our implementation;

$\mathbf{s}^i$ is the $i$-th speaker embedding in the pre-trained parameters $\boldsymbol{\mu}$, which learned from the training data, and $M$ is the number of speaker embeddings in the pertained model. Similar to Eq.9 introduced in the Sec.4.1, we can also use multiple independent speaker models to get a more robust guidance.

## 5 EXPERIMENTS

### 5.1 DATASET

We use three datasets in this paper; one to train the unsupervised score generation model, one for the ASR guidance model, and another for the speaker guidance model.

The first one is the LibriTTS-R (Koizumi et al., 2023) dataset. It is a quality-improved dataset derived from LibriTTS (Zen et al., 2019). The sampling rate is 24 kHz. LibriTTS-R provides the text annotation for each sample, but we do not use it during training (*unsupervised*). We use the 522-hour speech partition to train the unconditional score-based generative model. As said, such an unconditional score-based diffusion process is performed on 100-dimensional mel-scaled spectrum features (i.e., $F = 100$) extracted from the speech signal.

The dataset we use to train the ASR models is the WSJ SI-284 dataset. It contains about 80 hours of training data. The original dataset is 16kHz, and we upsample the data to 24 kHz and extract mel-spectrum features with the same parameter as that for LibriTTS-R.

The third dataset we use is the Voxceleb2 (Chung et al., 2018) dataset, which is used to train the speaker verification model. It includes 5994 speakers for model training, and the total amount of training data is about 2000 hours.

### 5.2 MODEL CONFIGURATIONS

We use the sub-variance preserving (sub-VP) introduced in Song et al. (2021) as the SDEs for unconditional score model training and set hyperparameters $\beta_{max} = 16.0$, $\beta_{min} = 0.1$. The NCSN++ (Song et al., 2021; Richter et al., 2023) network is used for the score model. The diffusion model is trained using Adam optimizer Kingma & Ba (2014) with an initial learning rate of $10^{-4}$. The learning rate decays by a factor of $0.97$ in every epoch. In each epoch, we optimize the model by 2000 steps with a batch of 2. The predictor-corrector sampler Song et al. (2021) is used to solve the reverse SDE with 100 discrete time step in the sampling stage. We apply the neural vocoder BigVGAN Lee et al. (2022), to resynthesize speech from the generated mel-spectrogram feature.

We trained 12 ASR models with the data perturbed by SDE on WSJ SI-284 as the ASR guidance models with the ESPNet toolkit (Watanabe et al., 2018). The 12 models include 4 different model sizes, each of them with 3 different byte pair encoding (BPE) (Sennrich et al., 2016). The detailed configurations are listed in the Appendix.A.1. All the ASR models have an AED structure and are optimized with joint CTC+attention loss (Kim et al., 2017).

For speaker guidance, we trained the speaker guidance models using the wespeaker toolkit Wang et al. (2024). We chose the ResNet34-based r-vector Zeinali et al. (2019) as the speaker embedding extractor and trained the systems on the SDE perturbed Voxceleb2 dataset following the wespeaker recipe[4]. To enhance the speaker guidance in the diffusion inference process, we trained two speaker guidance models by applying different random seeds.

---

[3]$\beta = 1000$ in our experiments.

[4]`https://github.com/wenet-e2e/wespeaker/tree/master/examples/voxceleb/v2`

## 5.3 EVALUATION METRICS

The generated speech is evaluated using both objective and subjective metrics. We generated 100 samples from the LibriTTS-R testing set for objective evaluation and 15 for subjective evaluation. In addition to the target text, we need to specify the total length that needs to be generated for each sample before sampling. We adopt the length of ground truth audio in the generation in most experiments unless otherwise stated. A detailed ablation study of target speech length will be conducted in Sec. 6.4.

**Objective evaluation**. We first report the word error rate (WER), which is tested on an ASR model trained on the Librispeech (Panayotov et al., 2015) dataset. The evaluation model is available online [5]. The second metric is UTMOS Saeki et al. (2022), a pseudo mean opinion score (MOS) predicted by a neural network. The third is the SpeechBERT (Chuang et al., 2020; Saeki et al., 2024) score, which measures the BERTScore (Zhang et al., 2019) for generated and reference speech with self-supervised dense speech features. The last metric is speaker similarity, which is used to evaluate the effect of speaker guidance. We extract the speaker embeddings using a publicly available speaker-id embedding model [6], and calculate the cosine similarity between the enrolment speech and generated speech. The UTMOS, SpeechBERT score, and speaker similarity are evaluated by the VERSA toolkit [7].

**Subjective evaluation.** We perform a human evaluation on the generated examples based on three criteria: N-MOS (naturalness Mean Opinion Score) for fluidity and naturalness, M-MOS (meaningfulness Mean Opinion Score) for meaningfulness and content quality, and S-MOS (Speaker Similarity Mean Opinion Score) for the speaker similarity between generated speech and the enrollment utterance. 20 listeners are asked to evaluate 15 utterances on a scale from 1 to 5. The instructions for subjective evaluations are provided in Appendix A.2.

## 6 RESULTS AND ANALYSIS

Table 1: Objective evaluation for different ASR guidance. The number of parameters of the unconditional SGM is 147.4 M. We also list the additional parameters in guidance ASR models.

| ASR Guidance ID | # total ASR param. (M) | WER(%) ↓ | UTMOS ↑ | SpeechBERT ↑ |
|---|---|---|---|---|
| {1} | 14.9 | 81.3 | 2.84 | 0.64 |
| {1, 2} | 29.8 | 39.9 | 3.18 | 0.69 |
| {1, 2, 3} | 44.8 | 24.4 | 3.28 | 0.71 |
| {1, · · · , 6} | 89.7 | 14.9 | 3.42 | 0.72 |
| {1, · · · , 9} | 147.2 | 11.1 | 3.44 | 0.73 |
| {1, · · · , 12} | 229.9 | 10.1 | 3.47 | 0.74 |
| Ground Truth | - | 3.3 | 4.15 | 1.00 |

## 6.1 EXPERIMENTAL RESULTS ON ASR GUIDANCE

We compare the effect of ASR guidance in Table.1. We trained the 12 ASR guidance model on 80-hour WSJ data, and they are identified by ID 1-12. The detailed configurations of them can be found in the Appendix A.1. We first tried to guide the unconditional SGM trained on LibriTTS-R with a single ASR, the results of which are listed in the first line of Table.1. We found that the speech generated by the guided diffusion by just one ASR system performed poorly in the objective evaluation. In contrast, if the WER is evaluated using the same ASR model used for guidance, it is near zero. That is because the generated speech data is optimized directly for the guidance ASR in the sampling process, similarly as it happens for white-box attacks (Wang et al., 2022). In other words, the guidance model can be easily fooled by low-quality samples generated by the guided sampling, especially as the data space $\mathbb{R}^{L \times F}$ is high-dimensional and sparse. A simple and

---

[5]https://huggingface.co/asapp/e_branchformer_librispeech
[6]https://huggingface.co/espnet/voxcelebs12_rawnet3
[7]https://github.com/shinjiwlab/versa/tree/main

straightforward approach that we adopt is to use multiple independent ASR models to provide more robust guidance, as described in Eq.9. The joint likelihood assessment of multiple ASR models during the sampling stage makes the proposed approach more robust and less prone to collapse to trivial solutions.

We identify the ASR guidance models trained on WSJ with ID $1 \sim 12$. Their details can be found in the Appendix.A.1. We gradually increased the number of guidance ASR models from 1 to 12, the WER can be significantly reduced from $81.3\%$ to $10.1\%$. It is worth noting that all the guidance ASR models are trained on the same 80-hour WSJ training set. Although we used guidance models up to 12, each model's parameter scale is deliberately kept relatively small for efficiency. When using 6 models as guidance, their total parameters (89.7M) do not exceed those of the unconditional score model (147.4M), and the word error rates can be reduced from $81.3\%$ to $14.9\%$.

## 6.2 EXPERIMENTAL RESULTS ON SPEAKER GUIDANCE

Table 2: Objective evaluation for speaker guidance. SIM is the speaker similarity between the generated speech and enrollment speech.

| Guidance | # total guide param. (M) | WER ↓ | SpeechBERT ↑ | UTMOS ↑ | SIM ↑ |
|---|---|---|---|---|---|
| ASR {1} | 14.9 | 81.3 | 0.64 | 2.84 | 0.16 |
| + 1 Spk. model | 21.6 | 83.1 | 0.62 | 2.86 | 0.29 |
| + 2 Spk. model | 28.3 | 80.0 | 0.63 | 2.87 | 0.37 |
| ASR {1,2,3} | 44.8 | 24.4 | 0.71 | 3.28 | 0.16 |
| + 1 Spk. model | 51.5 | 23.7 | 0.71 | 3.41 | 0.31 |
| + 2 Spk. model | 58.2 | 23.5 | 0.71 | 3.37 | 0.38 |
| ASR {1,..,6} | 89.7 | 14.9 | 0.72 | 3.42 | 0.15 |
| + 1 Spk. model | 96.4 | 14.8 | 0.73 | 3.45 | 0.29 |
| + 2 Spk. model | 103.1 | 14.9 | 0.73 | 3.48 | 0.38 |
| ASR {1, …, 12} | 229.9 | 10.1 | 0.74 | 3.47 | 0.12 |
| +1 Spk. model | 236.6 | 9.0 | 0.74 | 3.47 | 0.27 |
| +2 Spk. model | 243.3 | 10.3 | 0.74 | 3.46 | 0.34 |
| Ground Truth | - | 3.3 | 1.00 | 4.15 | 0.59 |

We apply the target speaker guidance by following Eq. 12 and Eq. 13. The enrollment speech $\mathbf{X}^c$ is randomly picked from other speech samples from the same speaker of the ground truth speech in the LibriTTS-R testing set. The speaker similarity evaluated between the ground truth and their enrollment speech is $0.59$. We compare the speaker guidance with one and two guidance speaker models. The results are listed in Table.2. In the speaker guidance experiments, we found that only using one speaker model as guidance can clearly improve speaker similarity while using two speaker models can improve it further. With the improvement of speaker similarity by introducing speaker guidance, most of the systems slightly reduce the WER, while some of them get a slightly worse WER. No obvious conflicts between the speaker and ASR guidance are observed, which is consistent with the independent assumption used in Eq.11 and Eq.12. When more ASR guidance models were used (12), speaker similarity improved less, which can be observed in the results of 12 ASR guidance.

## 6.3 RESULTS ON MEAN OPINION SCORE EVALUATIONS

We report the MOS evaluation results in Table 3. We get M-MOS and N-MOS scores around 3.9, which still has a gap between the TTS speech and the real speech ($\sim 4.9$). Relative comparisons between different guidance are consistent with those in Sec.6.1 and Sec.6.2. The 12-ASR guidance setup shows better M-MOS and N-MOS than the 6-ASR guidance but gets worse S-MOS than the latter one. However, both systems equipped with speaker guidance show significant S-MOS improvement when compared to the system without speaker guidance.

Table 3: Mean Opinion Score (MOS) for the Guided TTS.

| ASR Guidance | Speaker Guidance | M-MOS | N-MOS | S-MOS |
|---|---|---|---|---|
| $\{1, ..., 6\}$ | 2 models | 3.87 | 3.80 | 3.26 |
| $\{1, ..., 12\}$ | 2 models | 3.99 | 3.95 | 2.92 |
| $\{1, ..., 12\}$ | None | 3.92 | 3.86 | 1.42 |
| Ground Truth | | 4.90 | 4.92 | 4.11 |

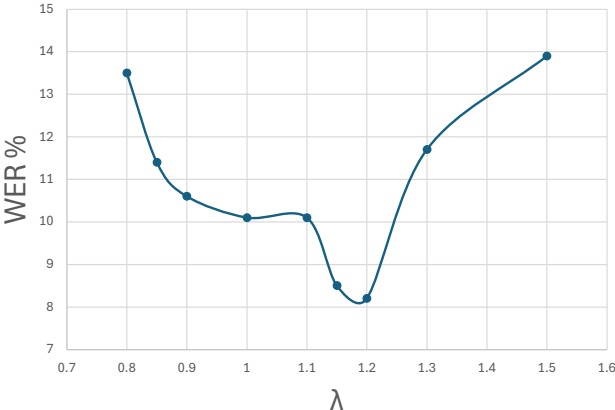

Figure 1: WER(%) w.r.t. the target speech length. $\lambda$ is the scaling factor to the ground truth length. Where $\lambda = 1$ is identical to the 12-ASR guidance in Table. 1.

## 6.4 ABLATION STUDIES ON SPEECH DURATION

In our proposed approach, the E2EASRs are leveraged to control speech generation for target text. E2EASRs learn an implicit alignment between the speech and text labels in their training stage. When using E2EASRs as guidance to control the TTS generation, we do not explicitly control the speed of speech or duration of phonemes. Our method only needs the desired total speech length, target text, and optional enrollment speech as input. In our previous experiments, we did not investigate the total length of the target speech in detail but simiply adopted the ground truth length as the target length. In real applications that do not have a ground truth length, an algorithm is needed to predict the target speech length from the text.

To understand the effect that the defects of the length prediction algorithm may have on the quality of TTS, we conducted an ablation experiment on the total length of speech. We scale the length of ground truth speech by a factor $\lambda$, and use it as the target length in the sampling. The guidance models are all the 12 WSJ E2EASR models. The curves of WER with respect to $\lambda$ are plotted in Fig 2.

As Fig.2 shows, the proposed methods are generally more sensitive to short speech length. If the total length of speech generation is shortened, the WER will increase. However, if the length of speech is longer than the ground truth in a reasonable range (from 1.0 to 1.5), the WER evaluation becomes on par or even better. A possible explanation for this phenomenon is that the LibriTT-R training data are well-segmented, and most of the onset and offset silence audio are clipped out. On the other hand, the WSJ ASR training data has more silence on both the onset and offset of the speech. So, the ground truth speech length in LibriTTS-R may be shorter for the distribution learned by the ASR guidance model. We give more detailed examples in the Appendix. A.3, the model is trying to generate onset and offset silence if the target length is set too long.

This finding can guide the design of length prediction algorithms in real applications. For example, the predictor can be biased to output a longer length than the ground truth.

## 7 DISCUSSIONS AND CONCLUSION

In this paper, we proposed to use E2EASR models to guide an unconditional score-based generative model (SGM) and enable TTS. The unconditional SGM can be trained on large-scare unlabeled speech data. We show that using ASR models trained only on 80-hour can guide the unconditional SGM to generate high-quality speech. Meanwhile, using speaker verification models as guidance, we can also conduct zero-shot TTS with a few seconds of the target speaker's enrollment speech. By utilizing the end-to-end training for the guidance ASR, We found that we can synthesize high-quality speech without using the phoneme duration model. An ablation study shows that our model is robust against the mismatch of total target speech length within a certain range.

The main limitations of our work and possible extensions include:

1) Multiple ASR guidance system are currently needed to generate satisfactory-quality speech, and our best system uses 12 ASR models for guidance. We use deliberately compact, small guidance ASR models ($\approx 15\,\mathrm{M}$ parameters), thus in part alleviating the computational overhead in the inference stage. Our future works will focus on reducing the number of guidance models. Possible solutions include optimizing the guidance ASR model against adversarial attacks to improve the robustness of guidance.

2) Although we disentangle the training data of the unconditional SGM and the guidance ASR model, the guidance model must be trained with the data perturbed by the diffusion SDE. This prevents arbitrary pre-trained models from being directly used as the guidance model. In the future, it is necessary to explore using guidance models that do not require data-perturbed training but e.g. only fine-tuning with perturbed data.

3) Our proposed E2EASR guidance for TTS does not require commonly used phoneme duration prediction models. We can generate speech with the input text and a total duration of the target length. In our experiments, we empirically set the latter to the length of the ground truth. In future work, the problem of generating natural speech robustly at a total input length needs to be explored.

4) This work focuses more on the utilization and optimization of guidance models but less on the design of unconditional SGM. An important research direction is how to design an unconditional SGM that is more effective when being guided.

5) Another potential extension for this work is cross-lingual zero-shot TTS. Using ASR models trained on a target language may be still be able to guide an unconditional SGM trained on single or multiple unlabeled speech data from other languages. This may solve the problem of lack of high-quality minority languages or dialects TTS annotation.

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

## A  APPENDIX

### A.1  DETAILS OF ASR GUIDANCE MODELS

A total of twelve ASR guidance models were trained on the WSJ dataset (referred to as WSJ-ASR 1 to 12). The WSJ-ASR models utilize a Transformer encoder-decoder architecture, with each model featuring 2048 hidden units and 4 attention heads.

All ASR guidance models were trained using the Adam optimizer (Kingma & Ba, 2014), with an initial learning rate of 0.001 and 5,000 warm-up steps. Each model was trained for 100 epochs. We employed a joint CTC-attention training framework (Kim et al., 2017), where the loss weights for the CTC and attention objectives were empirically set to 0.3 and 0.7, respectively. We use byte-pair encoding (BPE) tokens for all models. Further architectural details of the ASR guidance models are provided in Table 4.

### A.2  INSTRUCTIONS FOR SUBJECTIVE EVALUATIONS

N-MOS: Your task is to judge the **Naturalness** of the speech you hear in relation to the reference speech. Please concentrate on the **fluidity and naturality of the interaction as well as the expressiveness of the speakers regardless of meaning**.

M-MOS: Your task is to judge the **Meaningfulness** of the speech you hear in relation to the reference speech. Please focus on **whether the sequence of words is identical to the reference speech**.

S-MOS: Your task is to judge the **Speaker Similarity** of the speech you hear in relation to the reference speech. Please concentrate on the **speaker similarity regardless of speech quality, naturalness, and meaning**.

### A.3  ABLATION STUDIES ON TARGET LENGTH

Figure 2 shows two speech syntheses with a longer or shorter length than the ground truth length. Although our methods do not include an explicit duration model, they are still robust in generating

Table 4: Details of ASR models

|  | # token | # encoder layers | # decoder layers | # param.(M) |
|---|---|---|---|---|
| 1 | 100 | 6 | 3 | 14.9 |
| 2 | 200 | 6 | 3 | 14.9 |
| 3 | 300 | 6 | 3 | 15.0 |
| 4 | 100 | 6 | 3 | 14.9 |
| 5 | 200 | 6 | 3 | 14.9 |
| 6 | 300 | 6 | 3 | 15.0 |
| 7 | 100 | 8 | 4 | 19.1 |
| 8 | 200 | 8 | 4 | 19.2 |
| 9 | 300 | 8 | 4 | 19.2 |
| 10 | 100 | 12 | 6 | 27.5 |
| 11 | 200 | 12 | 6 | 27.6 |
| 12 | 300 | 12 | 6 | 27.7 |

speech based on the target length. For the longer length, it will try to speak slower and generate silence at the beginning and end of the speech; for the shorter length, it will try to talk faster. It is worth noting that our method has no control over whether to speak slower or add silence at the beginning or end of the audio when the target speech length is too long. The specific approach to control it needs to be studied in the future. Our current model is not good at producing perfect silence because the unconditional diffusion model's training data is well segmented and contains fewer data that start or end with a longer duration. If this problem is solved in future work, the model can be more robust against longer input speech lengths.

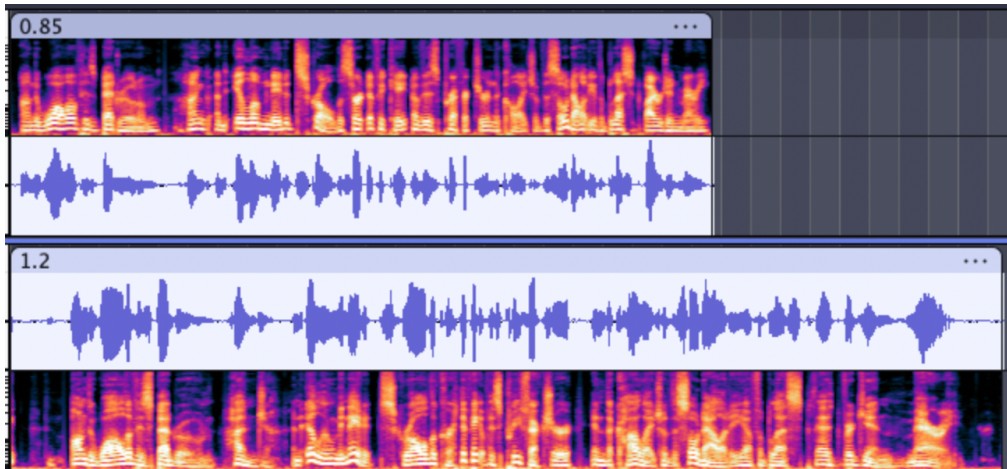

Figure 2: Speech synthesized with 1.2 times or 0.85 times length of the ground truth length.

