# OpenReview forum: "Speaking Guided by Listening: Unsupervised Text-to-Speech Generative Model Guided by End-to-End Speech Recognition"
_ICLR.cc/2025/Conference — ICLR 2025 Conference Withdrawn Submission_

### Official Review · Reviewer_bT6q · 2024-10-18

**Soundness:** 2
**Presentation:** 2
**Contribution:** 2
**Rating:** 3
**Confidence:** 4

**Summary:**

This paper proposes to use classifier guidance for diffusion-based speech synthesis where the classifier is an end-to-end ASR model.

**Strengths:**

The research direction is interesting and the use of classifier guidance in diffusion-based TTS models is not explored.

**Weaknesses:**

1. The contribution of this paper is limited. It applies existing techniques to diffusion-based TTS. The application might be novel but it is more suitable for a shorter paper (e.g. speech-specific conferences).
2. Key experiment missing - how does it compare to classifier-free guidance trained on the same LibriTTS data? Say using the same ASR system to transcribe the speech content and then use that as the input to train a classifier-free diffusion model?

**Questions:**

See weaknesses.

---

### Official Review · Reviewer_hPez · 2024-10-31

**Soundness:** 2
**Presentation:** 1
**Contribution:** 2
**Rating:** 3
**Confidence:** 5

**Summary:**

This study approaches TTS modeling differently, using unconditional speech generation and a separate ASR model for decomposition. Unlike similar previous studies (e.g., Guided-TTS 1,2), it employs a CTC-based classifier to guide speech, removing the need for a phoneme duration predictor. Additionally, a verification module for speaker guidance eliminates the need for speaker conditioning in speech generation.

**Strengths:**

- The authors propose a technique to guide speech with multiple ASR models to prevent poor guidance, improving pronunciation accuracy at the cost of increased parameters.

- They eliminated the need for phoneme-level alignment by training a CTC-based text classifier.

**Weaknesses:**

1. Despite guiding with multiple ASR models, pronunciation accuracy remains lower than GT in a large gap. If the proposed method significantly impacts pronunciation accuracy, it should ideally be compared to norm-based guidance used in similar research, such as Guided-TTS, which aimed to improve pronunciation accuracy.

2. Utilizing multiple ASR models inevitably increases computational costs. Calculating gradients in parallel consumes memory, while sequential calculations slow down inference speed.

3. The primary advantage of the authors’ model over previous studies is that it functions as an unconditional speech generation model without requiring either text or speaker conditions. However, the LibriTTS data used in the experiments allows easy extraction of speaker IDs or embeddings with existing open-source speaker verification model. To demonstrate the benefit of speaker guidance via a verification model, a comparison with a speaker-conditioned diffusion model using explicit labeling would have been beneficial.

4. Similarly, text guidance should be compared to conventional TTS structures.

5. Finally, no confidence intervals are provided for MOS measurements, which are essential.

**Questions:**

Points of interest or suggestions are outlined in the Weaknesses section.

---

### Official Review · Reviewer_8rTp · 2024-11-04

**Soundness:** 1
**Presentation:** 2
**Contribution:** 1
**Rating:** 3
**Confidence:** 4

**Summary:**

This paper presents a diffusion style text-to-speech system that uses a supervised ASR model to guide an unconditional speech diffusion model with the classifier-based guidance framework. The authors also show a speaker verification model can be applied similarly to control the generated voice.

**Strengths:**

* The authors showed the using multiple ASR model for guidance can improve the performance.
* The authors extend classifier guidance for speaker control

**Weaknesses:**

* The motivation of the proposed solution is unclear. This is not an unsupervised TTS model if supervised data is already used to train the ASR model that is required for running inference. For semi-supervised setup, what is the benefit of classifier guidance compared to pre-training - fine-tuning approach in [1]?
* There is very limited novelty. Guided-TTS already presented a similar TTS model that uses a phoneme predictor as the classifier guidance. The difference is this paper uses an E2E-ASR instead. It appears that the only new insight is using an ensemble of ASR model improves the performance.
* Does using more ASR models improve because ensemble model avoids adversarial samples (high likelihood but low quality) or simply because the guidance weight increases effectively? The authors should also present ablation results scanning through different guidance weights with a single model / fewer models.
* This paper did not compare with any prior work of similar setups and discuss what the benefits of the proposed methods are compared to other semi-supervised (Guided-TTS, SpeechFlow [1]) or unsupervised TTS systems [2].

[1] Generative pre-training for speech with flow matching
[2] Simple and Effective Unsupervised Speech Synthesis

**Questions:**

* What perturbation is used for the ASR model (line 182)? How much does the performance degrade if the ASR model is not trained on perturbed Xt?
* See other questions in the weakness section

---

### Official Review · Reviewer_WntG · 2024-11-09

**Soundness:** 2
**Presentation:** 3
**Contribution:** 3
**Rating:** 5
**Confidence:** 4

**Summary:**

This paper proposed a method to combine an unconditioned diffusion-based speech generation model with separated trained ASR models to achieve text-to-speech generation via classifier guidance. Importantly, the experiments showed that such a method doesn't work well with a single ASR model, because the trained diffusion model may be overfit to the ASR model used, and generates speech that sounds poorly but fools the ASR model. However, simply increasing the number of ASR guidance be improve the performance drastically.

**Strengths:**

The proposed method is simple and sound.

**Weaknesses:**

There are important questions left unaddressed. See questions section below.

**Questions:**

1. For the impact of the number of ASR guidance -- is that more because of this number, or the performance of the ASR model (in the paper it's just a tiny model trained with only 80 hours data)? Can you compare it to using a more sophisticated ASR model, for example, Whisper, which is publicly available off the shelf?

2. This paper does not address the duration for the speech to be generated, but instead, used the groundtruth duration. This makes the works incomplete as a TTS model. Sec 6.4 made preliminary inspection on the impact of the duration prediction with WER metrics, but does not touch other aspects such as MOS.

3. Table 1: The impact of the number of ASR guidance is drastic, and doesn't seem to saturate by 12. Why not further increase it in the experiments? BTW, would it be better to show column 1 as number of ASR guidance, instead of listing all the IDs (same for Table 2)?

---

### Note · Authors · 2024-11-28

I have read and agree with the venue's withdrawal policy on behalf of myself and my co-authors.